# Prevalence and distribution pattern of *Cryptosporidium* spp. among pre-weaned diarrheic calves in the Republic of Korea

**Dong-Hun Jang**[1], **Hyung-Chul Cho**[1], **Seung-Uk Shin**[1], **Eun-Mi Kim**[1], **Yu-Jin Park**[1], **Sunwoo Hwang**[1], **Jinho Park**[2], **Kyoung-Seong Choi**[1] *

**1** Department of Animal Science and Biotechnology, College of Ecology and Environmental Science, Kyungpook National University, Sangju, Republic of Korea, **2** College of Veterinary Medicine, Jeonbuk National University, Iksan, Republic of Korea

\* kschoi3@knu.ac.kr

**Data Availability Statement:** All relevant data are within the paper.

**Funding:** Kyoung-Seong Choi: This research was supported by the Korea Institute of Planning and

## Abstract

*Cryptosporidium* spp. are protozoan parasites that belong to subphylum apicomplexa and cause diarrhea in humans and animals worldwide. Data on the prevalence of *Cryptosporidium* spp. and its subtypes among calves in the Republic of Korea (KOR) are sparse. Hence, our study aimed to investigate the prevalence and association between the age of calf and the identified *Cryptosporidium* spp. and to determine the genotypes/subtypes of *Cryptosporidium* spp. in pre-weaned calves with diarrhea in the KOR. A total of 460 diarrheic fecal samples were collected from calves aged 1−60 days and screened for *Cryptosporidium* spp. by the 18S rRNA gene. Species identification was determined using the sequencing analysis of the 18S rRNA gene, and *C. parvum*-positive samples were subtyped via the sequence analysis of the 60-kDa glycoprotein (*gp60*) gene. Sequence analysis based on the 18S rRNA gene revealed the presence of three *Cryptosporidium* spp., namely, *C. parvum* ($n = 72$), *C. ryanae* ($n = 12$), and *C. bovis* ($n = 2$). Co-infection by these species was not observed. The infection rate was the highest in calves aged 11−20 days (26.1%, 95% CI 17.1−35.1), whereas the lowest rate was observed in calves aged 21−30 days (7.7%, 95% CI 0.0−16.1). The prevalence of *C. parvum* was detected exclusively in calves aged ≤20 days, and the highest infection rate of *C. ryanae* was seen in calves ≥31 days of age. The occurrence of *C. parvum* ($\chi^2 = 25.300$, $P = 0.000$) and *C. ryanae* ($\chi^2 = 18.020$, $P = 0.001$) was significantly associated with the age of the calves. Eleven different subtypes of the IIa family that belonging to *C. parvum* were recognized via the sequence analyses of the *gp60* gene. Except for two (IIaA18G3R1 and IIaA15G2R1) subtypes, nine subtypes were first identified in calves with diarrhea in the KOR. IIaA18G3R1 was the most frequently detected subtype (72.2% of calves), followed by IIaA17G3R1 (5.6%), IIaA15G2R1 (4.2%), IIaA19G4R1 (4.2%), IIaA16G4R1 (2.8%), IIaA17G4R1 (2.8%), IIaA19G3R (2.8%), IIaA14G1R1 (1.4%), IIaA14G3R1 (1.4%), IIaA15G1R1 (1.4%), and IIaA19G1R1 (1.4%) These results suggest that the prevalence of *Cryptosporidium* spp. is significantly associated with calf age. Furthermore, the findings demonstrate the high genetic diversity of *C. parvum* and the widespread occurrence of zoonotic *C. parvum* in pre-weaned calves.

Evaluation for Technology in Food, Agriculture, and Forestry (IPET) (Grant No. 321016-01-1-HD020). The funders had no role in study design, data collection and analysis, decision to publish, or preparation of the manuscript.

**Competing interests:** The authors have declared that they no competing interests exit.

Hence, calves are a potential source of zoonotic transmission with considerable public health implications.

## Introduction

*Cryptosporidium* spp. are protozoan parasites that cause mild-to-severe diarrhea in humans and a wide range of animals [1]. Infections with these parasites occur via the fecal–oral route either by direct contact with infected animals or by the ingestion of infective oocysts from contaminated water or food [2–5]. To date, 40 *Cryptosporidium* spp. have been described [6], and among them, four species, namely, *C. andersoni*, *C. bovis*, *C. parvum*, and *C. ryanae*, have been identified in cattle. The distribution of these species is known to vary according to age [4, 7]. In particular, *C. parvum* is one of the most important pathogens causing diarrhea in neonatal calves worldwide and leads to severe economic losses owing to poor growth, decreased productivity, and even death [8]. Moreover, *C. parvum* is the major pathogenic species that affects humans [9, 10]. Unlike *C. parvum*, *C. bovis*, and *C. ryanae* usually infect post-weaned calves and yearlings without causing illness, and *C. andersoni* is mainly found in adult cattle [11–13]. The pathogenicity of *C. bovis*, and *C. ryanae* in post-weaned calves has not been established [9]. The oocysts of *C. parvum*, *C. bovis*, and *C. ryanae* are similar in size and shape. While *C. ryanae* is smaller than the others and requires molecular methods for its determination [14, 15], *C. andersoni* is larger in size and infects the abomasum [16].

According to the subtyping of *C. parvum* based on sequence analysis of the 60-kDa glycoprotein (*gp*60) gene, IIa and IId subtypes have been detected in both humans and calves and can cause zoonotic cryptosporidiosis [17]. The IIa subtype is mostly identified in calves, and IIaA15G2R1 is the predominant subtype [7] globally, including the Republic of Korea (KOR) [18]. The IId subtype is usually found in lambs and goat kids [4, 19] and has been described in calves in some countries such as Sweden, Turkey, Egypt, and China [20–23]. To date, most investigations of cryptosporidiosis in calves caused by *C. parvum* have focused on the IIa subtype in most countries. However, there are a few studies on *C. parvum* subtypes in calves in the KOR [18, 24].

*Cryptosporidium parvum* infects the intestinal mucosa and accounts for over 90% of *Cryptosporidium* infections in neonatal calves [23]. In contrast, in pre-weaned calves, the prevalence of *C. bovis* and *C. ryanae* and their effects on causing diarrhea remain unclear. Several studies have reported that *C. bovis* and *C. ryanae* are present in pre-weaned calves [23, 25, 26] and that *C. ryanae* infections are particularly associated with moderate diarrhea in pre-weaned calves [23]. However, little is known about the association between *C. bovis* and diarrhea. In addition, a previous study has indicated the high prevalence of *C. bovis* and *C. ryanae* in hemorrhagic diarrhea in the KOR [24]. Nevertheless, the pathogenicity of these organisms is still unclear.

So far, for the identification of *Cryptosporidium* spp., a nested polymerase chain reaction (PCR) technique based on the SSU rRNA gene has been the most widely used method [27]. However, in the present study, a conventional PCR method using species-specific primers was used [24]. Although the amplification had a short fragment compared with a previous method, this PCR technique enabled the differentiation between *C. bovis* and *C. ryanae*. Therefore, this study aimed to investigate the prevalence of *Cryptosporidium* spp. using species-specific primers in pre-weaned calves with diarrhea and to evaluate the association between the age of calf and the identified *Cryptosporidium* spp. Furthermore, we intended to determine the genotype of *Cryptosporidium* spp. and subtyping of *C. parvum* in calves in the KOR and to assess the significance of calves as a source of human infections.

## Materials and methods

### Ethics statement

All animal procedures were conducted according to ethical guidelines for the use of animal samples, and were approved by the Jeonbuk National University (Institutional Animal Care and Use Committee Decision No. CBNU 2020–052). All procedures and possible consequences were explained to the managers of the surveyed farm, and written consent was obtained.

### Sample collection

Between August 2019 and August 2020, fresh fecal samples were collected directly from the rectum of 460 diarrheic pre-weaned calves (up to 60 days of age) by an experienced veterinarian using sterile plastic gloves in 11 different farms located in the KOR. The samples were placed in labeled sterile plastic tubes and transported to the Animal Immunology Laboratory of Kyungpook National University in a cooler with ice packs. Upon arrival, sampling date, age, animal identification number, and fecal consistency (pasty, loose, watery, or hemorrhagic) were recorded for each animal. The collected feces were mostly pasty or loose. Prior to DNA extraction, all feces were stored at 4°C for no more than 2 days without the additional treatment of preservation. The fecal samples were divided according to age as follows; 1−10 days ($n$ = 271), 11−20 days ($n$ = 92), 21−30 days ($n$ = 39), and ≥31 days ($n$ = 58). No microscopic examination was performed for the detection of oocysts.

### DNA extraction, molecular analysis, and sequencing

DNA was extracted from 200 mg of each fecal sample using the QIAamp Fast DNA Stool Mini Kit (Qiagen, Hilden, Germany) according to the manufacturer's instructions. In brief, samples were suspended in lysis buffer, followed by boiling at 70°C for 5 min. Next, the inhibitors provided in the kit were added to the solution to remove substances that can degrade DNA and inhibit downstream enzymatic reactions. Supernatants were subsequently transferred into a tube containing proteinase K and then heated at 70°C for 10 min. A final volume of 200 μL of each DNA sample was then stored at −20°C until PCR amplification. The identification of *Cryptosporidium* spp. was first tested using the 18S rRNA gene [28]. Samples that yielded positive results for *Cryptosporidium* spp. via the sequence analysis were further screened to detect the four species using species-specific primers [24]. Positive samples for *C. parvum* were retested using the 60-kDa glycoprotein (*gp60*) gene to determine its subtype [4], whereas positive samples for *C. bovis*/*C. ryanae* were differentiated by sequence analysis. The subtypes of *gp60* were named based on the repeated number of TCA (A), TCG (G), and ACATCA (R), as described previously [29]. All positive PCR products were purified using the AccuPower PCR Purification Kit (Bioneer, Daejeon, KOR) and employed for direct sequencing (Macrogen, Daejeon, KOR). The nucleotide sequences obtained in this study were analyzed using BioEdit (version 7.2.5) and compared with the reference sequences using the Basic Local Alignment Search Tool available at the National Center for Biotechnology Information database. As the sequences of *C. bovis* and *C. ryanae* are highly similar, all amplified samples were differentiated by comparing the sequences between the two species. To determine the subtype of *C. parvum* as well as the genotypes of *C. bovis* and *C. ryanae*, nucleotide sequences were aligned using ClustalX and then analyzed via direct comparison with reference sequences from GenBank. In this study, only samples showing a good sequencing result were considered positive for each *Cryptosporidium* spp. All nucleotide sequences generated in this study were deposited in the

**Table 1. Prevalence and distribution of *Cryptosporidium* species according to age group in pre-weaned calves.**

| Age (days) | Sample size | No. of positive (%) | 95% CI | *Cryptosporidium* species (No.) | | |
|---|---|---|---|---|---|---|
| | | | | *C. parvum* | *C. ryanae* | *C. bovis* |
| 1−10 | 271 | 53 (19.6%) | 14.8−24.3 | 49 | 3 | 1 |
| 11−20 | 92 | 24 (26.1%) | 17.1−35.1 | 23 | 1 | 0 |
| 21−30 | 39 | 3 (7.7%) | 0.0−16.1 | 0 | 3 | 0 |
| 31−60 | 58 | 6 (10.3%) | 2.5−18.2 | 0 | 5 | 1 |
| Total | 460 | 86 (18.7%) | 15.1−22.3 | 72 | 12 | 2 |

GenBank database with appropriate accession numbers (18S rRNA: MZ736386−MZ736399; *gp60*: MZ736314−MZ736385).

## Statistical analysis

Statistical analysis was performed using SPSS Statistics 26 software package for Windows (SPSS Inc, Chicago, IL, USA). Chi-square test was used to determine the association between the prevalence of each species and age. Moreover, multinomial logistic regression analysis was used to determine any associations between the subtypes of *C. parvum* and age. A *p*-value of less than 0.05 was considered statistically significant.

# Results

## Prevalence of *Cryptosporidium* spp.

Among the 460 diarrheic fecal samples examined, 86 (18.7%) were positive for *Cryptosporidium* spp. on PCR analysis and sequencing based on the 18S rRNA gene. Three *Cryptosporidium* spp. were identified in pre-weaned Korean native calves (Table 1). No *C. andersoni* was detected in this study. Of these, *C. parvum* (15.7%, 72/460) was the most detected, followed by *C. ryanae* (2.6%, 12/460) and *C. bovis* (0.4%, 2/460). Co-infection of these species was not observed. The prevalence of the three *Cryptosporidium* spp. was compared according to the age groups. As shown in Table 1, the infection rate of *Cryptosporidium* spp. was highest in calves aged 11−20 days (26.1%, 95% CI 17.1−35.1), whereas the lowest infection rate was observed in calves aged 21−30 days (7.7%, 95% CI 0.0−16.1). All three *Cryptosporidium* spp. were detected only in calves aged 1−10 days (Table 1). The association between *Cryptosporidium* spp. and age-distribution was investigated. Interestingly, the identified *Cryptosporidium* spp. varied according to the age of the calves. *C. parvum* infection was detected exclusively in calves ≤20 days of age (Table 2). The prevalence peaked at the age of 11−20 days and decreased rapidly thereafter (Table 2). *C. parvum* infection was significantly associated with the age of the calves ($\chi^2$ = 25.300, *P* = 0.000). Unlike *C. parvum*, *C. ryanae* was found in all age groups, and the highest infection rate was observed at ≥31 days of age (Table 2). *C. ryanae* infection

**Table 2. Distribution of *Cryptosporidium* species in pre-weaned Korean native calves according to age group.**

| Age (days) | Frequency of *C. parvum* positivity (%) | $\chi^2$ (*P*-value) | Frequency of *C. ryanae* positivity (%) | $\chi^2$ (*P*-value) | Frequency of *C. bovis* positivity (%) | $\chi^2$ (*P*-value) |
|---|---|---|---|---|---|---|
| 1−10 | 49/271 (18.1%) | 25.300 (0.000) | 3/271 (1.1%) | 16.020 (0.001) | 1/271 (0.4%) | 2.824 (0.419) |
| 11−20 | 23/92 (25.0%) | | 1/92 (1.1%) | | 0 | |
| 21−30 | 0 | | 3/39 (7.7%) | | 0 | |
| 31−60 (Ref.) | 0 | | 5/58 (8.6%) | | 1/58 (1.7%) | |

also had a significant age-related distribution ($\chi^2$ = 18.020, *P* = 0.001). In contrast, *C. bovis* was detected only in two calves aged 10 days and 35 days, and there was no statistical significance in the age-related distribution (*P* = 0.590).

### Distribution of *Cryptosporidium* spp. and *C. parvum* subtypes

All 72 *C. parvum*-positive samples were successfully amplified and subtyped by sequence analysis of the *gp60* gene. A total of 11 different subtypes belonging to the family IIa were identified (Table 3). Subtype family IId was not detected. The distinction of each subtype within the IIa was in the number of trinucleotide region of TCA and TGA repeats (i.e., had one copy of sequence ACATCA immediately after the trinucleotide repeats). As shown in Table 3, in pre-weaned Korean native calves, the most frequently detected subtype was IIaA18G3R1 (72.2%), followed by IIaA17G3R1 (5.6%), and then IIaA15G2R1 (4.2%) and IIaA19G4R1 (4.2%). Other subtypes, namely, IIaA14G1R1 (1.4%), IIaA14G3R1 (1.4%), IIaA15G1R1 (1.4%), IIaA16G4R1 (2.8%), IIaA17G4R1 (2.8%), IIaA19G1R1 (1.4%), and IIaA19G3R1 (2.8%) were also identified. Except for the IIaA18G3R1, no statistical correlation was found between calf age and a specific subtype (Table 3). IIaA19G4R1 was observed only in calves aged 1−10 days, whereas IIaA17G3R1 was found exclusively in calves aged 11−20 days. Several more subtypes were found in calves aged 1−10 days (Table 3). The most predominant subtype, IIaA18G3R1, was seen in all ages.

Based on the 18S rRNA gene, 14 (12 *C. ryanae* and 2 *C. bovis*) sequences were obtained and compared with the published literature. Twelve sequences of *C. ryanae* showed 95.1%−100% similarity with each other. The *C. ryanae* sequences shared 95.7%−100% identity with those found in Austria, China, India, Thailand, and Japan. Two sequences of *C. bovis* shared 94.1% similarity. These sequences demonstrated 95.5%−96.2% identity with those identified previously in the KOR and had 91.9%−96.2% homology with those from Austria, USA, Japan, and China. Interestingly, differences in nucleotides between *C. ryanae* and *C. bovis* were observed. As shown in Fig 1, the nucleotides in the six positions, i.e., 440, 460, 464−466, and 470, were different between the two species.

### Discussion

*Cryptosporidium*, along with rotavirus, has been well recognized as the main pathogen causing diarrhea in neonatal calves worldwide [30]. Our findings established the prevalence of

**Table 3. Distribution of *Cryptosporidium parvum* subtype according to age group.**

| *gp60* subtypes | Age groups (days) | | No. of positive calves | *P*-value |
|---|---|---|---|---|
| | 1−10 | 11−20 | | |
| IIaA18G3R1 | 36 | 16 | 52 (72.2%) | 0.000 |
| IIaA17G3R1 | 1 | 3 | 4 (5.6%) | 0.753 |
| IIaA15G2R1 | 3 | 0 | 3 (4.2%) | 0.785 |
| IIaA19G4R1 | 3 | 0 | 3 (4.2%) | 0.785 |
| IIaA16G4R1 | 1 | 1 | 2 (2.8%) | 0.823 |
| IIaA17G4R1 | 1 | 1 | 2 (2.8%) | 0.823 |
| IIaA19G3R1 | 0 | 2 | 2 (2.8%) | 0.677 |
| IIaA14G1R1 | 1 | 0 | 1 (1.4%) | 0.874 |
| IIaA14G3R1 | 1 | 0 | 1 (1.4%) | 0.874 |
| IIaA15G1R1 | 1 | 0 | 1 (1.4%) | 0.874 |
| IIaA19G1R1 | 1 | 0 | 1 (1.4%) | 0.874 |
| Total | 46 | 26 | 72 | |

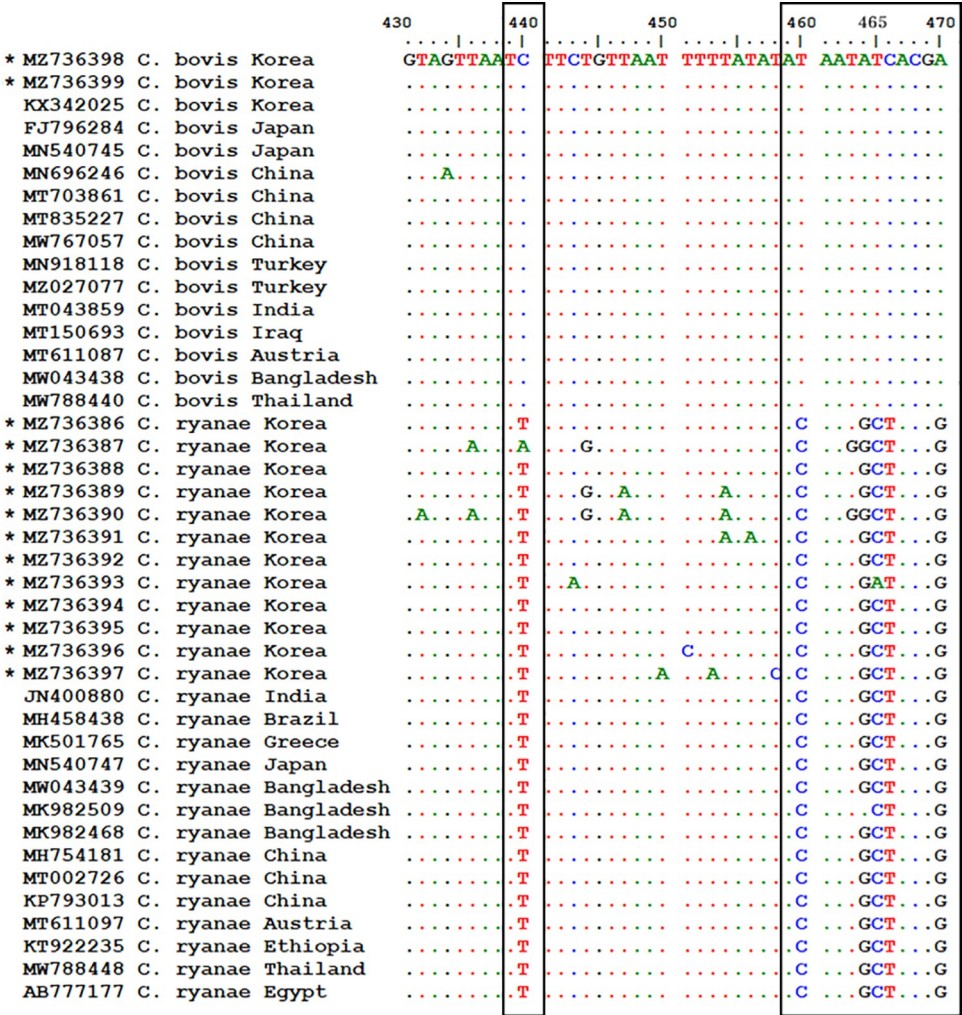

**Fig 1. Sequence comparisons between *C. bovis* and *C. ryanae* for the partial18S rRNA gene from Korean sequences obtained in this study and reference strains.** Six nucleotide differences at 440, 460, 464–466, and 470 are shown. An asterisk indicates sequences obtained in this study.

*Cryptosporidium* spp. in pre-weaned diarrheic calves according to age, and the presence of various zoonotic subtypes of *C. parvum* in the KOR were identified. In the present study, the overall prevalence of *Cryptosporidium* spp. was found to be 18.7%, which is higher than that reported previously in the KOR [18, 24, 31]. These variations could be explained by the age of the animals, time of sample collection, and the differences in geographical location. However, the percentage of *Cryptosporidium* spp.-positive samples found in our study was lower than that reported in other countries such as Germany (88.9%), Japan (83.8%), China (38.4%), Italy (38.8%), Colombia (26.6%), Argentina (22.5%), and Estonia (22.6%) [25, 32–37].

In this study, the presence of three *Cryptosporidium* spp. in pre-weaned Korean native calves was ascertained: *C. bovis*, *C. parvum*, and *C. ryanae*. Of them, *C. parvum* was the most predominant species in the KOR. This finding agrees with the results observed in several other countries [7, 25, 33, 36, 38, 39]. Most studies have proven that *C. parvum* mainly infects calves up to 1 month of age [33, 40–43]. The results of the present study demonstrated that *C. parvum* was detected only in calves aged ≤20 days, and the infection rate was the highest in calves aged

11–20 days. This observation is consistent with a previous study performed by our group [18]. According to our findings, *C. parvum* was detected in calves aged ≤20 days. It is considered that calves in this age group are susceptible to *C. parvum* infection owing to their immature immune system [44]. In addition, it is well known that young calves can become infected with *C. parvum* and begin shedding the oocysts soon after birth [45–47]. This could be associated with cow-to-calf transmission. Several studies have reported that the possible source of infection in calves is transmission at birth from their mothers [48, 49]. However, at present, we do not have exact information on whether these calves were immediately removed from their mothers after birth, but the possibility of contamination via exposure to mother's feces or the surroundings should be considered. Moreover, *C. parvum* is known to cause watery diarrhea [23, 30]. In this study, the number of animals with watery feces was small; hence, the association with diarrhea was not evaluated. Although we were not able to compare the occurrence of *C. parvum* with the diarrhea status, *C. parvum* was found to be the causative agent of diarrhea in young calves. Our results suggest that *C. parvum* infection is attributed to the significant age-related distribution (*P* = 0.000). Consequently, *C. parvum* was strongly associated with diarrhea in calves aged ≤20 days.

*Cryptosporidium ryanae* was the second most frequently detected species in pre-weaned Korean native calves. In general, *C. ryanae* is often found in post-weaned calves [15]. The results revealed that *C. ryanae* was detected in all age groups and that its occurrence increased with age. In particular, the infection rate of *C. ryanae* showed a low prevalence in calves aged <20 days, whereas it was rather high in calves aged ≥31 days (Table 2). The prevalence of *C. ryanae* found in this study was similar to that of a previous study performed in the KOR [24]. Our observation confirmed that *C. ryanae* has an age-associated distribution, similar to *C. parvum*. A recent study has reported that *C. ryanae* was common in pre-weaned as well as post-weaned calves and that the infection was associated with the occurrence of moderate diarrhea in pre-weaned calves [23]. In contrast, other studies have shown that *C. ryanae* was not associated with diarrhea [26, 39, 50]. So far, the pathogenicity of *C. ryanae* is controversial. A previous study conducted in the KOR demonstrated that although it is not a single infection, the prevalence of *C. ryanae* was significantly high in hemorrhagic diarrhea [24]. We could not arrive at a conclusion regarding the correlation with diarrhea since the number of *C. ryanae*-positive samples from diarrheic calves was small. Hence, *C. ryanae* infection may cause diarrhea in calves ≥21 days of age and should be considered as a causative agent of diarrhea in this age group. Further studies are necessary to clarify the pathogenicity of *C. ryanae* in pre-weaned calves.

We found that the prevalence of *C. bovis* was the lowest in pre-weaned Korean native calves. This observation is contradictory to the results reported by several studies in which *C. bovis* was the dominant species in pre-weaned calves [20, 47, 51–53]. In this study, *C. bovis* was detected only in two calves aged 10 and 35 days. Several studies have stated that *C. bovis* is common in 2–3-week-old calves [42, 50]. However, our result signified that *C. bovis* was not detected in this age (Table 1). Cai et al. mentioned that *C. bovis* usually appears after weaning and that the infection can last weeks or months and contribute to the small increase in *Cryptosporidium* infection rates soon after weaning [26]. This observation may also explain the low prevalence of *C. bovis* in the present study. To date, information on the prevalence and clinical signs of *C. bovis* infection in both pre-weaned and post-weaned calves is very limited in the KOR. *C. bovis* could have probably been considered to be less important than *C. parvum* and therefore overlooked as an etiological agent of diarrhea in calves. Moreover, the results revealed that infection by *C. bovis*, unlike the two other species, was not age-related. Most importantly, the involvement of *C. bovis* in diarrhea remains unclear. Unlike *C. ryanae*, many studies have suggested that *C. bovis* was associated with diarrhea [23, 26, 39, 54]. However,

infection by *C. bovis*/*C. ryanae* may lead to clinical signs owing to the presence of *C. parvum* [33]. Therefore, the prevalence and pathogenicity of *C. bovis* in pre-weaned and post-weaned calves must be investigated through large-scale epidemiological surveys.

*C. parvum* IIa family is common in humans as well as calves and is considered potentially zoonotic. To date, three *C. parvum* subtypes have been detected in calves in the KOR [18, 24], whereas one subtype (IIaA16G3R1) was not found in this study. In addition to the two subtypes (IIaA15G2R1 and IIaA18G3R1) described above, nine other subtypes (IIaA14G1R1, IIaA14G3R1, IIaA15G1R1, IIaA16G4R1, IIaA17G3R1, IIaA17G4R1, IIaA19G1R1, IIaA19G3R1, and IIaA19G4R1) that have not previously been detected in the KOR were identified for the first time, showing the presence of high genetic diversity. Among them, IIaA18G3R1 was most commonly found in pre-weaned Korean native calves with diarrhea. This result is inconsistent with that of a previous study in which IIaA15G2R1 was shown as the predominant subtype [18]. This difference could be attributed to the fact that in the previous study, both normal and diarrheic feces were used and that IIaA15G2R1 was detected regardless of diarrhea [18]. Other variations are due to the differences in the season of sampling, regions, the number of samples, and herd management. IIaA15G2R1 has been known as the most prevalent *C. parvum* subtype infecting humans and cattle in many countries [7, 34, 55–59] and has also been detected in calves without diarrhea [18, 33, 60]. There seems to be no relationship between the subtype and diarrhea. In the present study, IIaA15G2R1 was detected only in three calves with diarrhea and was the third frequent subtype along with IIaA19G4R1.

Here, IIaA18G3R1 was the dominant subtype that accounted for 72.2% of *C. parvum*-infected pre-weaned Korean native calves and was the frequent cause of human cryptosporidiosis, besides being reported in calves and foals [61–66]. The second common subtype in the KOR, IIaA17G3R1, has been found in calves and humans in several countries [67–71]. IIaA19G4R1 was the third frequent subtype identified in the pre-weaned Korean native calves and was also detected in small ruminants and fish as well as humans and calves [61, 70, 72–74]. Interestingly, all sequences belonging to the IIaA19G4R1 subtype were identical to those reported from other countries previously. These subtypes are considered to be the most common ones in calves in the KOR.

The other seven subtypes were also identified in pre-weaned Korean native calves with diarrhea, but their prevalence was relatively low. Subtypes IIaA14G1R1, IIaA14G3R1, and IIaA15G1R1 were each detected in one calf. IIaA14G1R1 was identified in calves, goat kids, and humans [7, 12, 17, 19, 25, 34, 57, 58]. IIaA14G3R1 was found in humans, calf, lambs, and fresh molluscan shellfish [19, 25, 75, 76]. IIaA15G1R1 has been reported in humans [29, 57, 58, 77, 78] as well as in cattle and goat kids [22, 79–81]. Subtypes IIaA16G4R1 and IIaA17G4R1 were each found in two calves in the current study. Unlike the other subtypes, IIaA16G4R1 has so far been noted only in neonatal calf with diarrhea [82], which is consistent with our findings. Subtype IIaA16G4R1 has not yet been detected in humans; however, the possibility that this may represent a significant health risk cannot be excluded. The IIaA17G4R1 subtype has been identified in humans, cattle, and goats [32, 34, 65, 76, 82, 83] and has also been detected in diarrheic calves [32]. Finally, subtypes IIaA19G1R1 and IIaA19G3R1 have each been identified in one calf. IIaA19G1R1 has been reported in humans, cattle, and sheep [36, 58, 69, 84–86]. IIaA19G3R1 has been identified in humans, cattle, and deer [66, 87–90]. To the best of our knowledge, this is the first study to report the presence of various subtypes in pre-weaned calves in the KOR.

To detect *C. bovis* and *C. ryanae*, 18S rRNA and heat-shock protein 70 genes are generally used [15]. According to sequence analysis of the 18S rRNA gene, *C. bovis* and *C. ryanae* showed ≥99% identity, and it is not always possible to differentiate between them by PCR [91, 92]. However, in this study, we used only the 18S rRNA gene. Even without phylogenetic

analysis, the difference between the two species could be confirmed via sequence analysis. At the six nucleotide positions of 440, 460, 464–466, and 470, *C. bovis* had C, T, A, T, C, and A, whereas *C. ryanae* had T, C, G, C, T, and G, respectively. These positions are representative markers that distinguish *C. ryanae* from *C. bovis*. Our results suggest that these two species can be discerned using the 18S rRNA gene.

## Conclusion

Our results confirm the presence of three *Cryptosporidium* spp. in pre-weaned calves with diarrhea: *C. bovis*, *C. parvum*, and *C. ryanae*. *C. parvum* was found to be the dominant species in young calves in the KOR. The occurrence of *C. ryanae* and *C. parvum*, but not *C. bovis*, in pre-weaned Korean native calves was significantly related to age; the prevalence of *C. parvum* decreased with age, whereas that of *C. ryanae* increased with age. The most frequently detected subtype in calves with diarrhea was IIaA18G3R1, which was responsible for zoonotic transmission. This is the first report to identify nine potentially zoonotic subtypes belonging to the family IIa, which have not previously been reported in cattle in the KOR. This study establishes the high genetic diversity of *C. parvum* in diarrheic calves and the widespread distribution of zoonotic *C. parvum* in the KOR. Therefore, the results emphasize that young calves may be a potential source of infection and may serve as an important zoonotic reservoir for human cryptosporidiosis [47, 49].

## Author Contributions

**Conceptualization:** Jinho Park, Kyoung-Seong Choi.

**Data curation:** Jinho Park, Kyoung-Seong Choi.

**Formal analysis:** Dong-Hun Jang, Hyung-Chul Cho, Seung-Uk Shin, Eun-Mi Kim, Yu-Jin Park, Sunwoo Hwang.

**Funding acquisition:** Kyoung-Seong Choi.

**Methodology:** Dong-Hun Jang, Hyung-Chul Cho, Seung-Uk Shin, Eun-Mi Kim, Yu-Jin Park, Sunwoo Hwang.

**Supervision:** Kyoung-Seong Choi.

**Writing – original draft:** Kyoung-Seong Choi.

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
