## [Decision Letter · Decision Letter 0]

6 Oct 2021

PONE-D-21-26307Prevalence and distribution of Cryptosporidium spp. among diarrheic calves in the Republic of KoreaPLOS ONE

Dear Dr. Kyoung-Seong Choi,

Thank you for submitting your manuscript to PLOS ONE. After careful consideration, we feel that it has merit but does not fully meet PLOS ONE’s publication criteria as it currently stands. Therefore, we invite you to submit a revised version of the manuscript that addresses the points raised during the review process.

We look forward to receiving your revised manuscript.

Kind regards,

Saeed El-Ashram

Academic Editor

PLOS ONE

Journal Requirements:

Reviewers' comments:

Reviewer's Responses to Questions

**Comments to the Author**

1. Is the manuscript technically sound, and do the data support the conclusions?

Reviewer #1: Partly

Reviewer #2: Partly

Reviewer #3: Yes

Reviewer #4: Yes

2. Has the statistical analysis been performed appropriately and rigorously? 

Reviewer #1: Yes

Reviewer #2: Yes

Reviewer #3: Yes

Reviewer #4: Yes

3. Have the authors made all data underlying the findings in their manuscript fully available?

Reviewer #1: Yes

Reviewer #2: No

Reviewer #3: Yes

Reviewer #4: Yes

4. Is the manuscript presented in an intelligible fashion and written in standard English?

Reviewer #1: Yes

Reviewer #2: Yes

Reviewer #3: Yes

Reviewer #4: Yes

5. Review Comments to the Author

Reviewer #1: Dear Author,

I can say that I really like the subject of your article.

However, there is some confusion. What is the method by which you specify subtypes? According to which literature did you identify these subtypes?

I have not seen such a method in the literature number 24 that you use on this subject.

You should provide information about this in your article.

Kind regards

Reviewer #2: The current MS presented Cryptospordiosis distribution among calves in ROK. The main issue of this work is the methods in which the authors described it in a briefly way. As PLOS ONE has broad base of readers, so, authors should be mentioned the methods used in something of details. Extraction of DNA from fecal samples needs specific precautions, authors need to mention how they manage these samples, add preservative or not, what about the inhibitors from feces? Also, I can't understand how the authors differentiate between C. ryanae and C. parvum? How did the authors decide that this is C. ryanae and other one is C. bovis then they did sequence and then differentiate between them?

-The introduction needs a paragraph about importance and improvements that were done in the molecular identifications of Cryptosporidium.

- In the discussion the authors need to discusses why the age below 20 days calves are more susceptible to C. parvum?

Reviewer #3: Dear authors,

Thank you for this interesting work.

The manuscript is well written.

Here you are my comments/ remarks for your consideration.

Title

Line2

Should be changed: “Prevalence and distribution pattern of Cryptosporidium spp. among pre-weaned diarrheic calves in the Republic of Korea”

Abstract

L24: Add Cryptosporidium spp. are protozoan parasites that belong to subphylum apicomplexa and cause diarrhea in humans and animals worldwide

L42: Illustrate that family IIa belongs to C. parvum

L45, 46: add the percentage of the rest subtypes that you found .

Key words

L52 Add : spp. to Cryptosporidium,

Materials and Methods

L93-L131 The Materials and Methods should be written in more details.

L110: Why did not you collect an equal number of samples in different age groups ?

Thus you can get more accurate prevalence and actual association between age and Cryptosporidium spp. distribution.

Results and discussion

Results and discussion were written in a good details.

Reviewer #4: Dear Author

This is a generally well written manuscript with minor grammatical errors. Just a few comments or corrections:

1. Correlations are mentioned in the results (line 170) but not described in the statistical analysis.

2. Consider moving the sentence in line 174 to the discussion as it is not a description of the results.

3. Line 216 statement on prevalence of C. ryanae in calves is ambiguous please revise.

4. Line 296 change "sequencing analysis" to "sequence analyses".

5. Line 297 add the word "respectively after the G at the end of the sentence. Line 313 add a recommendation.

Thank you and well done on this work.

6. PLOS authors have the option to publish the peer review history of their article (what does this mean?). If published, this will include your full peer review and any attached files.

Reviewer #1: No

Reviewer #2: No

Reviewer #3: No

Reviewer #4: No

---

## [Author Response · Author response to Decision Letter 0]

20 Oct 2021

Reviewer #1: Dear Author,

I can say that I really like the subject of your article.

However, there is some confusion. What is the method by which you specify subtypes? According to which literature did you identify these subtypes?

I have not seen such a method in the literature number 24 that you use on this subject.

You should provide information about this in your article.

Kind regards

Response: We are sorry for the confusion and agree with this comment. As the reviewer’s comment, we have provided them. Please see lines 131-133 and 466-468.

 

Reviewer #2: The current MS presented Cryptospordiosis distribution among calves in ROK. The main issue of this work is the methods in which the authors described it in a briefly way. As PLOS ONE has broad base of readers, so, authors should be mentioned the methods used in something of details. Extraction of DNA from fecal samples needs specific precautions, authors need to mention how they manage these samples, add preservative or not, what about the inhibitors from feces? Also, I can't understand how the authors differentiate between C. ryanae and C. parvum? How did the authors decide that this is C. ryanae and other one is C. bovis then they did sequence and then differentiate between them?

Response: We agree with this comment and understand the reviewer’s concern. Unfortunately, we did not examine oocysts from each feces, because we are not expert in this field. However, we used PCR method to detect Cryptosporidium species. All feces were stored at 4�C without additional treatment of preservation and were used for DNA extraction. We extracted directly DNA from feces using the QIAamp Fast DNA Stool Mini Kit (Qiagen) according to the manufacturer's instructions. We have provided the methods as the reviewer’s suggestion. Please see lines 121-126. 

In addition, to detect Cryptosporidium species, we first used the 18S rRNA gene and sequenced all positive samples. By sequencing analysis, samples that yielded positive results for Cryptosporidium spp. were further screened to identify the species using species-specific primers. Positive samples for C. parvum was re-tested using the 60-kDa glycoprotein (gp60) gene to determine its subtype. To differentiate between C. bovis and C. ryanae, amplified sequences were compared and confirmed. According to the results of sequence, two species were different from each other. Please see lines 127-131, and 138-139.

-The introduction needs a paragraph about importance and improvements that were done in the molecular identifications of Cryptosporidium.

Response: In this study, we utilized the partial 18S rRNA gene, not SSU rRNA gene. In general, SSU rRNA gene is based on a nest PCR method. As you know, nested PCR has some problems such as contamination risks owing to multiple rounds of DNA amplification and concomitant DNA manipulation steps. To exclude these problems, we used conventional PCR method which have a short fragment. This method can easily be used to detect Cryptosporidium spp. We have provided them. Please see lines 87-91.

- In the discussion the authors need to discusses why the age below 20 days calves are more susceptible to C. parvum?

Response: We agree with this comment. At this point, we cannot make a conclusion about this, but it could be related to immune system of calves. In addition, it has been reported that C. parvum can be transmitted from cow to calf. Please see lines 219-227.

 

Reviewer #3: Dear authors,

Thank you for this interesting work.

The manuscript is well written.

Here you are my comments/ remarks for your consideration.

Title

Line2

Should be changed: “Prevalence and distribution pattern of Cryptosporidium spp. among pre-weaned diarrheic calves in the Republic of Korea”

Response: We have changed as the reviewer’s suggestion. Please see lines 2-3.

Abstract

L24: Add Cryptosporidium spp. are protozoan parasites that belong to subphylum apicomplexa and cause diarrhea in humans and animals worldwide

Response: We have modified as the reviewer’s suggestion. Please see lines 22-23.

L42: Illustrate that family IIa belongs to C. parvum

Response: We have modified as the reviewer’s suggestion. Please see lines 39-40.

L45, 46: add the percentage of the rest subtypes that you found .

Response: We have provided as the reviewer’s suggestion. Please see lines 43-45.

Key words

L52 Add : spp. to Cryptosporidium,

Response: We have modified as the reviewer’s suggestion. Please see line 51.

Materials and Methods

L93-L131 The Materials and Methods should be written in more details.

Response: We agree with this comment. As the reviewer’s comment, we have provided them. Please see lines 114-115, 117, 121-133, and 138-140.

L110: Why did not you collect an equal number of samples in different age groups ?

Thus you can get more accurate prevalence and actual association between age and Cryptosporidium spp. distribution.

Response: We understand the reviewer’s concern and agree with this comment. Our lab is focused on calf diarrhea, especially neonates. As you know, calf diarrhea occurs mainly less than 1 month old. Because most of the samples requested to be tested by our lab are under 30-day-old and feces were less in the age group over 30 days of age. For this reason, currently, it is difficult to compare the association between age group and distribution of Cryptosporidium spp. We are going to try to compare the prevalence and association between age and Cryptosporidium spp., if the number of samples is similar in all age groups. Please understand our situation. Thank you for your comment.

Results and discussion

Results and discussion were written in a good details.

 

Reviewer #4: Dear Author

This is a generally well written manuscript with minor grammatical errors. Just a few comments or corrections:

1. Correlations are mentioned in the results (line 170) but not described in the statistical analysis.

Response: We agree with this comment. We have provided. Please see lines 186-187 and Table 3. 

2. Consider moving the sentence in line 174 to the discussion as it is not a description of the results.

Response: We agree with this comment. We have removed it and added in discussion. Please see lines 311-312.

3. Line 216 statement on prevalence of C. ryanae in calves is ambiguous please revise.

Response: We are sorry for the confusion and have revised. Please see lines 236-238.

4. Line 296 change "sequencing analysis" to "sequence analyses".

Response: We have changed as the reviewer’s suggestion. Please see lines 317-318.

5. Line 297 add the word "respectively after the G at the end of the sentence. Line 313 add a recommendation.

Response: We have modified as the reviewer’s suggestion and added some references. Please see line 319 and 335.

Thank you and well done on this work.

 

Reviewer comment

1. Line 111. No microscopic examination was performed. How did you know for sure that the stools could be infective? Was it a coincidence?

Response: We agree with this comment and understand the reviewer’s concern. We were not able to examine the oocytes because we are not expert in this field. So, we tested all fecal samples for detection of Cryptosporidium infection. 

2. Line 114. DNA was extracted from 200 mg. Specify what pretreatment was done to break up the oocyst wall.

Response: We understand the reviewer’s concern. Because we did not test the oocysts and did not perform the step to break up the oocysts wall. To extract DNA, we directly used feces according to the manufacturer’s instructions. Please see lines 114-115, 117, and 121-126. 

3. Line 119. As far as I know, C.bovis/ryanae cannot be differentiated with this gene region. Clarification on the primers used should be clarified.

Response: We agree with this comment. We have provided the reference of species-specific primers which we use. Please see line 129. 

4. Line 126. Since the base of sequences of C. bovis and C. ryanae are similar, all positive samples of the 18S rRNA were separated by comparing the sequences. Which method or according to which technique was this process done?

Response: We agree with this comment and understand the reviewer’s concern. We first used the 18S rRNA gene to detect Cryptosporidium species and positive samples for Cryptosporidium species by sequencing analysis were screened to identify the four species using species-specific primers. Finally, positive samples for C. bovis/C. ryanae were differentiated by sequence analysis and species was confirmed based on the sequence analysis. According to our results, these two species had differences in nucleotide sequences. 

5. Line 142-143. If C. andersoni was not found, please specify.

Response: We have provided. Please see lines 158-159.

---

## [Decision Letter · Decision Letter 1]

27 Oct 2021

Prevalence and distribution pattern of Cryptosporidium spp. among pre-weaned diarrheic calves in the Republic of Korea

PONE-D-21-26307R1

Dear Dr. Kyoung-Seong Choi,

We’re pleased to inform you that your manuscript has been judged scientifically suitable for publication and will be formally accepted for publication once it meets all outstanding technical requirements.

Kind regards,

Saeed El-Ashram

Academic Editor

PLOS ONE

Additional Editor Comments (optional):

Thank you very much for all of your hard work and dedication. I, the academic editor, welcome your future submissions on behalf of the PLoS ONE editorial members.

Reviewers' comments:

Reviewer's Responses to Questions

**Comments to the Author**

1. If the authors have adequately addressed your comments raised in a previous round of review and you feel that this manuscript is now acceptable for publication, you may indicate that here to bypass the “Comments to the Author” section, enter your conflict of interest statement in the “Confidential to Editor” section, and submit your "Accept" recommendation.

Reviewer #1: (No Response)

Reviewer #2: All comments have been addressed

Reviewer #3: All comments have been addressed

Reviewer #4: All comments have been addressed

2. Is the manuscript technically sound, and do the data support the conclusions?

Reviewer #1: Yes

Reviewer #2: Yes

Reviewer #3: Yes

Reviewer #4: Yes

3. Has the statistical analysis been performed appropriately and rigorously? 

Reviewer #1: Yes

Reviewer #2: Yes

Reviewer #3: Yes

Reviewer #4: Yes

4. Have the authors made all data underlying the findings in their manuscript fully available?

Reviewer #1: Yes

Reviewer #2: Yes

Reviewer #3: Yes

Reviewer #4: Yes

5. Is the manuscript presented in an intelligible fashion and written in standard English?

Reviewer #1: Yes

Reviewer #2: Yes

Reviewer #3: Yes

Reviewer #4: Yes

6. Review Comments to the Author

Reviewer #1: Accept

It can be published in this form

best wishes

Accept

It can be published in this form

best wishes

Reviewer #2: The authors addressed all comments and corrections that recommended by the reviewer. The manuscript could be accepted in this form

Reviewer #3: Dear authors,

Thank you for performing all the modifications i requested in the manuscript.

Well done and good effort.

With my best wishes

Reviewer #4: Dear Authors

Thanks for the thorough revision of the article and addressing all the reviewer comments. The manuscript is reading well and I recommended it to be accepted for publication.

7. PLOS authors have the option to publish the peer review history of their article (what does this mean?). If published, this will include your full peer review and any attached files.

Reviewer #1: No

Reviewer #2: **Yes: **Shawky M Aboelhadid

Reviewer #3: No

Reviewer #4: No

---

## [Editor Report · Acceptance letter]

5 Nov 2021

PONE-D-21-26307R1 

Prevalence and distribution pattern of *Cryptosporidium* spp. among pre-weaned diarrheic calves in the Republic of Korea 

Dear Dr. Choi:

I'm pleased to inform you that your manuscript has been deemed suitable for publication in PLOS ONE. Congratulations! Your manuscript is now with our production department. 

Kind regards, 

on behalf of

Professor Saeed El-Ashram 

Academic Editor

PLOS ONE